



**Title:**

Large-scale biospheric drought response intensifies linearly with drought duration

**Authors:**

René Orth[1]*, Georgia Destouni[2], Martin Jung[1], Markus Reichstein[1]

**Affiliations:**

[1] Department of Biogeochemical Integration, Max Planck Institute for Biogeochemistry, D-07745
Jena, Germany

[2] Department of Physical Geography, Bolin Centre for Climate Research, Stockholm University,
SE-10691 Stockholm, Sweden

* Corresponding author. E-mail: rene.orth@bgc-jena.mpg.de

**Abstract:**

Soil moisture droughts have comprehensive implications for terrestrial ecosystems. Here we study
accumulated impacts of the strongest observed droughts on vegetation. The results show that
drought duration, the time during which surface soil moisture is below seasonal average, is a key

diagnostic variable for predicting drought-integrated changes in (i) gross primary productivity, (ii)
evapotranspiration, (iii) vegetation greenness, and (iv) crop yields. Drought-integrated anomalies in
these vegetation-related variables scale linearly with drought duration with a slope depending on
climate. In arid regions, the slope is steep such that vegetation drought response intensifies with
drought duration, whereas in humid regions, it is small such that drought impacts on vegetation are

weak even for long droughts. These emergent large-scale linearities are not well captured by state-
of-the-art hydrological, land surface and vegetation models. Overall, the linear relationship of
drought duration versus vegetation response and crop yield reductions can serve as model
benchmark, and support drought impact interpretation and prediction.



## 1. Introduction

Drought has complex and potentially severe impacts on the terrestrial biosphere *(1,2,3,4)*. In particular, it affects the vegetation and can thereby reduce or even reverse carbon uptake from the atmosphere *(2)*, increase (heat wave) temperatures through reduced evaporative cooling *(5)*, increase wildfire activity *(6)*, and reduce food production *(7)*. These multifaceted drought effects on vegetation are relevant for economy and society, as well as for natural ecosystems. However, these

effects are complex *(3,8,9)*, with the drought response of plants (partly) non-linearly depending on various factors. These comprise, e.g., vegetation characteristics, such as root depth, leaf area and plant physiology, soil characteristics, such as water holding capacity, and hydrological and terrain characteristics, which in turn affect groundwater level and thereby also soil moisture conditions above it *(10,11)*. Moreover, drought history can also play a role through legacy effects *(12)*. The

interplay of these factors in determining vegetation responses to drought is not yet well understood, in particular over large spatial scales and with respect to different and changing climate conditions. Since recently, modern Earth observation through satellites and ground station networks in combination with radiative transfer modeling and/or upscaling approaches provide unprecedented large-scale datasets. Using such datasets it has become possible to identify dominant connections

between key observed variables during droughts, such as a strong and fast relationship between soil moisture and runoff deficits, emerging at large scale across hydrological catchments and climate zones of Europe *(13)*. Following and expanding this investigation pathway, the present study compiles and analyzes worldwide data, revealing simple and useful relationship(s) that integrate the complex large-scale vegetation response to droughts of different magnitude.

## 2. Data and Methods

Drought in this study is determined through surface soil moisture deficits, a simple and widely used drought indicator that is directly related to vegetation-accessible water availability *(1)*. For this purpose we use ESA CCI soil moisture data *(14)*. Further, in order to characterize the meteorological drought forcing, we employ precipitation from ERA-Interim *(15)* and net radiation

data from the SRB and CERES datasets. Moreover, to infer the vegetation drought response, we consider data for gross primary productivity (GPP,) and evapotranspiration (ET) from the

FLUXCOM-RS dataset *(16)*, and normalized differential vegetation index data (NDVI ) from the GIMMS3g dataset *(17)*. All employed datasets are derived from observations, and provide global coverage (see supporting information for details). We employ satellite-derived datasets where

available, namely for surface soil moisture, net radiation, and NDVI. For robust drought-effect assessment, these are complemented by precipitation, GPP and ET datasets, obtained from upscaled/interpolated site observations.

In addition to observational data, we use state-of-the-art modelled datasets. In particular, we consider (surface) soil moisture and ET from the GLEAM dataset *(18)* and from 6 models from the

Earth2Observe model ensemble (version 1, *(19)*, see also supporting information), which provide these two required variables. GLEAM is a land surface model that assimilates observations of soil moisture, vegetation optical depth, and snow water equivalent. The Earth2Observe ensemble includes ordinary land surface models as well as hydrological models. These models provide estimates of surface soil moisture, typically representing the top 2-10 cm (exact values are model-

dependent *(19)*), and can as such be compared with the satellite-derived product. In addition, the models provide total column soil moisture (representing typically 1-5 m; exact depths are model-dependent), which is used to assess the impact of considered soil moisture depth on our results. Besides these models, we further consider an ensemble of land surface and vegetation models from the TRENDY project (version 3, *(20, 21)*, see also supporting information). These models provide

ET and total column soil moisture. All model simulations considered here are uncoupled and forced with observation-based gridded meteorological data.

Constrained by concurrent availability of different required data streams, we consider the time period 2001-2015, half-degree spatial resolution, and half-monthly temporal resolution of the data for the observation-based analyses (see supporting information). While we use the same temporal

and spatial resolution for the model-based analyses, the time period considered in this context is slightly shifted, 1998-2012. This is because the Earth2Observe simulations do only extent until 2012. All time series are de-trended prior to further analyses, with trends determined using a 3-year moving average window. To study the biospheric drought response, we focus on droughts that peak during the local growing season. This is determined for each location (half-degree grid cell) as the 5

consecutive months with highest multi-year average NDVI, or ET in the case of the model-based

analyses.

We consider for each grid cell the strongest drought in terms of surface soil moisture within the

study period 2001-2015. This drought is identified in three steps: (i) we compute soil moisture

anomalies by removing the mean seasonal cycle from the actual time series; (ii) we determine the

driest anomaly in each year's growing season; and (iii) we select the drought with the greatest dry

anomaly out of the total 15 peak anomalies over the whole time-period. The duration of each

drought is then the period before and after drought peak, during which the soil moisture anomalies

are negative, i.e. when soil moisture is drier than the seasonal mean. Note that our drought

definition therefore does not include an indication for vegetation water stress; furthermore, rain

events may occur during diagnosed drought periods.

Anomalies are also computed for the other investigated variables, in the same way as for soil

moisture. To enable direct comparison of anomalies across variables, and across observations and

models, we perform a standardization by dividing all anomaly values with a characteristic

variability value. This is computed for each variable and each grid cell as the standard deviation

across *all* half-monthly growing-season values. This way, all anomalies discussed and illustrated in

this study are scaled by and expressed as inter-annual standard deviations.

Whenever vegetation-related anomalies are integrated over the course of particular droughts, the

integration is performed across a time window of 8 months. It starts 3 months before the half-

monthly period denoting drought peak, and finishes 4.5 months after the peak period. Not using the

specific actual duration of each drought for the integrations ensures to avoid spurious correlations

between drought duration and the drought-integrated vegetation response. Also for this purpose, the

above-described standardization is performed after the time-integration.

Finally, to characterize the climate, and to measure the relative roles of water- versus energy-

limitation for the water use by the vegetation, we compute an aridity index (Figure S1). This index

is derived by dividing the mean net radiation over the entire study period by the corresponding

precipitation mean.

## 3. Results

### 3.1 Global vegetation drought response

The global GPP response to the respective strongest grid-cell drought during the study period 2001-

2015 is displayed in Figure 1. Strongest negative anomalies are found across central North

America, eastern South America, southern Africa, and Australia. Note that these are normalized

anomalies; especially in very dry regions with low inter-annual vegetation variability, such as inner

Australia, the actual absolute anomalies are comparatively low. In contrast, we find positive GPP

anomalies across eastern China, northern Europe, central Russia and eastern North America, even

though the magnitude is overall smaller compared with the aforementioned negative anomalies. In

these regions, GPP anomalies are mostly insensitive to soil moisture drought but rather induced by

co-variations of dry soil moisture with other, in this more relevant GPP-limiting factors such as

radiation and temperature. Averaging the results across grid cells with similar aridity reveals a

strong dependency of the vegetation drought response on climate. This is the first main result of our

study; whereas anomalies are small in energy-limited conditions (aridity index < 1, i.e. more

precipitation than (equivalent) net radiation), they increase markedly for increasingly water-limited

conditions (aridity index > 1).

This aridity finding is consistent with results in *(22)*, and is mechanistically explainable by more

water being available as deeper soil water and groundwater in wetter regions. Further, this greater

water amount is also (more) accessible to the vegetation because the fraction of tree cover is higher

in wetter regions, implying deeper rooting systems. Accessing these water reservoirs can help

vegetation to bridge surface-soil drought conditions, while also benefitting from a surplus in net

radiation which is often associated with less precipitation *(23)*. Similar results as for GPP are also obtained for ET and NDVI (see Figures S2 and S3 in the supporting information), illustrating the

robustness of these findings.

### 3.2 Time evolution of drought and biospheric response

The evolution of drought across climate regions is analyzed by averaging data for each investigated variable across grid cells with similar aridity. The results of this composite approach *(24)* are displayed in Figure 2 for all meteorological forcing and biospheric response variables. The

precipitation deficits during drought buildup are commonly accompanied by a net radiation surplus. They jointly lead to soil moisture deficits. The comparatively large soil moisture anomalies result from our drought definition based on driest soil moisture anomalies. Only in the driest considered climate, no net radiation surplus is found. This might have to do with drought-induced albedo changes, which enhance the outgoing radiation. Interestingly, the peak vegetation responses are

delayed and occur after drought peak. This is consistent with site- and/or time-specific findings in earlier studies, analyzing particular drought events *(24-25)*. Aside from plant-physiological reasons, this can be explained with the pre-peak radiation surplus which tends to enhance vegetation functioning. By contrast, in the post-peak period, with both soil water deficit and radiation deficit, the vegetation functioning is decreased. The radiation deficit follows from the recovery

precipitation and the associated clouds that occur by definition after drought peak.

While the drought forcing shown in Figure 2a-c is comparable in regions with similar aridity, the vegetation drought response changes strongly as aridity exceeds 2. This non-linear response is consistent with findings in Figure 1. Finally, the GPP and NDVI signals are similar, illustrating robustness in observed vegetation response to drought across these different vegetation-related

variables and associated data products.

### 3.3 Drought duration shaping the biospheric drought response

In Figure 3 we analyze the role of drought duration (i.e. the time period with below-normal soil moisture). Drought duration has no systematic influence on the vegetation drought response in wet

areas (aridity index < 1), where GPP anomalies are comparatively small anyway. By contrast, the

emerging linearity between the drought-integrated GPP anomalies and the mean drought duration

with increasing slope towards drier conditions is another main result of this study. The slope does

not increase further between dry and very dry regions (aridity index > 4) as already the shortest

droughts lead to negative impacts due to limited (ground)water availability. The relatively large

inter-quartile range underlying the relationships shown in Figure 3 is likely due to the considerable

aridity condition variety within each considered aridity class that spans across a factor of 2. The

range also illustrates that other processes and conditions than just aridity and drought duration

contribute to the vegetation drought responses locally. These results are not sensitive to the chosen

drought definition; using the longest growing-season drought duration instead of the strongest half-

monthly soil moisture anomaly to determine the strongest drought at each grid cell, we obtain

similar linearity relationships (Figure S4). Further, as the choice of an 8-month time period for

integrating the vegetation drought response is necessarily arbitrary, we repeat the analysis from

Figure 3 with an integration period of 6 months and find very similar results (Figure S5).

Overall, these findings indicate that in addition to a region's mean aridity, drought duration is a key

diagnostic variable for characterizing the large-scale vegetation drought response, and consequently

also for inferring drought impact on the land-atmosphere exchanges of carbon and water. While the

relevance of drought duration has been recognized in previous studies *(26, 27)*, the simple linear

relationships identified here are an essential new step for straightforward representation and

advancement in understanding of drought impacts on vegetation, e.g., comparatively between

different historic time periods *(28)*, and associated ecosystem functioning and land-atmosphere

exchanges. Drought duration as a main diagnostic variable integrates different interacting factors on

vegetation functioning during drought. These include higher (lower) general and drought-initial soil

moisture levels in wetter (drier) climate, in which shorter (longer) droughts can develop, while also

water stresses are smaller (greater) and induce less (more) severe drought effects on vegetation.

The emergent linearity between vegetation response and drought duration is not trivial, given the

complex interacting processes contributing to biospheric drought responses *(1,3)*. This complexity

is, for example, illustrated by the delayed peak in the vegetation drought response in Figure 2d-e. Further research is needed to better understand why and how such simple large-scale relationships can capture the interplay of various small(er)-scale processes.

Figure S6 compares the power of drought duration to infer the large-scale GPP response to drought
with that of several alternative controls. The results confirm the role of drought duration as a simple and efficient prediction measure for biospheric drought impacts in semi-humid to arid regions (aridity index > 1), for which significant slopes are found (Figure 3). Other common drought description metrics fail to achieve similar explanatory power in these climate regions, including the number of consecutive dry days, which was proposed as a preferred drought index (in addition to
the soil moisture anomalies used to derive drought duration in this study) by the IPCC special report on extremes *(1)*.

### 3.4 Modelled versus observed vegetation drought response

While the large-scale vegetation response to drought duration was analyzed with GPP data in the previous sections, in Figure 4 we additionally consider ET and NDVI as alternative observation-
based variables, which also indicate the functioning of the vegetation. Overall, similar relationships are found for the 3 variables; this highlights the significance of the emerging linear pattern in summarizing various influences contributing to the biospheric drought response. However, in semi-humid climate (0.5 < aridity index < 2) the ET drought response differs somewhat from the NDVI and GPP responses, possibly due to changes in water use efficiency. Further, the NDVI drought
response is slightly less pronounced than those of GPP and ET in very dry regions (aridity index > 4).

In a further step, we evaluate the vegetation drought response from several state-of-the-art hydrological and land surface models in relation to the observation-based results. Note that a different time period is used in for the model-based analyses, 1998-2012 instead of 2001-2015.
While we cannot exclude an impact of this period shift on our conclusions, we can assume that it is



minimal as the observational and modelled time periods are of the same length, and they largely overlap.

In particular, we compare the state-of-the-art GLEAM model dataset with simulation results from the Earth2Observe model comparison project (see Section 2). In general, the modelled ET

responses to drought are overestimated in wet climate and underestimated in dry climate compared with the observations-based relationships. This result implies relatively low sensitivity to climate in the modelled vegetation drought response. The sensitivity is slightly higher for GLEAM than for the Earth2Observe models, leading to generally better GLEAM agreement with the observation-based relationships. Interestingly, the models capture the observed linearity in the vegetation

drought response only for short-to-medium drought durations. As such, in dry climate they fail to capture the further intensification of the ET drought response towards droughts longer than 6 months. The individual model results are broadly similar (see Table S1 in the supporting information), with a spread comparable to the inter-quartile range of the observation-based ET relationship.

In order to test the role of surface versus total-column soil moisture, we also re-compute Figure 4 with root-zone (GLEAM) and total-column soil moisture (Earth2Observe models). The results in Figure S7 (see Table S2 in the supporting information for individual model results) show a slightly weaker ET response to deeper soil moisture drought than to surface soil moisture drought. Overall, there is remarkable similarity across the drought response relationships for both soil moisture

depths, indicating relatively small soil-moisture depth impact on our results. This finding suggests that, while plant water availability is actually determined by deep(er) soil moisture, surface soil moisture is a reasonable proxy for meaningful estimation of the drought duration-vegetation response relationships studied here. In addition to the models used above, we also consider TRENDY models that only provide total soil moisture (see Section 2). The results found for these

models confirm the results of the Earth2Observe models; the TRENDY models generally do not capture the differences in the drought response relationships for different climates. Also the spread

across the drought response patterns of the TRENDY models is comparable to that of the Earth2Observe models.

Overall, the difficulties of models to capture the linearity between vegetation drought response and drought duration emerging from observations likely arise from the complex interplay of several small-scale processes leading to the large-scale relationships. Further model development efforts are required to improve simulated drought responses; the emergent linear relationships identified in this study can serve as a straightforward guideline and constrain in this context.

### 3.5 Drought duration and food production

The global vegetation drought responses emerging in the previous sections for GPP, ET, and NDVI should also be reflected in crop yields, with high social relevance. As crop yield data with consistent format and quality is only available across Europe, we correspondingly focus in the crop yield analysis in this section on Europe. Specifically, we analyze agricultural yield anomalies

averaged across 5 common crops (see supporting information), in the strongest drought year for various European countries. In the grid-cell analyses above, the year of the strongest drought has been determined at each grid cell through the strongest half-monthly soil moisture anomaly. While the strongest droughts therefore might occur in different years across the grid cells of a country, we select the year in which most of the strongest grid-cell droughts occur as the country-based drought

year (see supporting information, *(29)*). The drought duration in this country-based drought year is then determined as the mean across all grid-cell-based drought durations, weighted by the fraction of agricultural area in each grid cell *(29)*.

As shown in Figure 5, we find that, in addition to the drought-integrated GPP and NDVI anomalies, also the agricultural yield anomalies in drought years are linearly related to drought duration. Short

droughts can even be beneficial for food production, due to the associated net radiation surplus. Significantly different linear regression slopes (t-test, 5% level) are found for countries with and without large-scale irrigation. Countries without irrigation exhibit a steeper line slope and a higher explained fraction of variance (0.65 versus 0.25) than countries with irrigation, where the added

irrigation water tends to mitigate drought impacts, as reflected from the associated less steep line

slope. These differences are well in line with the contrast seen between arid and humid regions in

Figures 3 and 4. Overall, these results highlight the important socio-economic relevance of drought

duration as a key diagnostic variable for predicting vegetation drought response and associated

crop-yield anomalies.

## 4. Conclusions

The identified large-scale, aridity-dependent linearity in biosphere responses to drought has

important practical implications, especially as it is found globally and robustly across different

ecosystem-response variables. Drought duration as a key diagnostic variable in this context is: (i)

straightforward to measure and monitor, and (ii) efficient for representation and comparative

understanding of observed/interpreted vegetation responses to droughts *(28)*, as well as for

anticipation and planning for adaptation to impacts on agricultural crop yields of possible/projected

forthcoming drought years. This diagnostic also enables us to infer associated implications for

water and carbon cycling, and consequently also for atmospheric feedbacks. Such knowledge can

complement existing drought monitors *(30)* and support efficient irrigation efforts *(31)*. Moreover,

the identified linear relationships can serve as constraints that inform future model developments;

such observation-based references are required to improve modelled vegetation responses to

drought, which are currently largely insufficient. These model improvements can in turn also

contribute to improve weather forecasting through a more accurate representation of (drought-

related) water and carbon fluxes on land *(32)*.

Caveats of our analysis include, firstly, that observation-based global soil moisture is only available

for the surface soil, as microwave remote sensing only penetrates into the upper few centimeters of

the soil. In fact, the vegetation drought response rather depends on the root-zone soil moisture,

where the depth and extent of the root zone is also species-dependent. Nevertheless, our

conclusions are still valid due to the close soil hydraulics links between soil moisture at the surface

and in deeper soil. At daily time scales, surface soil moisture has been reported to reflect the



moisture dynamics across the top 10-20 centimeters *(33)*, and this depth is likely even greater at the monthly-seasonal time scales considered in this study, thereby capturing (at least part of) the actual root zones of many plant species. Moreover, we have tested the impact of using surface versus total-column soil moisture in our model analyses, finding only minor differences in the results (Figures 4 and S7). Secondly, due to limited observation data availability, we could in this study

only consider the strongest drought over 15 years. Hence the investigated droughts represent relatively weak extreme events, and it remains unclear if and how this affects the (strength) of the emergent linear relationships found in our study. Nevertheless, in some grid cells, droughts with return periods clearly exceeding 15 years occurred during the study period, for example in 2003 and 2015 in Europe *(34, 35)*. While we may have only captured a few very extreme droughts,

future research is needed to revisit our analysis with longer observational records capturing more extreme droughts. Thirdly, when analyzing the link between the vegetation drought response and drought duration it is inevitable that both variables are assessed over (partly) overlapping time periods. To avoid introducing a spurious relationship in this context, we use in this study a constant time window for the integration of the vegetation drought response (8 months, and comparatively

also 6 months), independent of the actual diagnosed drought duration. Further, the results obtained for the independent country-wise anomalies of yearly crop yield confirm the linearity resulting from the grid-cell-based analyses using the 8-month (or 6-month) window.

Finally, while we have found aridity and drought duration as main controls of the vegetation drought response at large spatial scales (climate regions), this is not necessarily the case at smaller

scales. In fact, the spread around the moving average relationships shown in Figures 3 and 4 suggests more drivers at play. These may include vegetation types, soil characteristics, and/or legacy effects. These drivers can intensify or dampen the local vegetation drought response compared with the large-scale response induced by the identified large-scale controls,

Overall, our results highlight an important role of climate (aridity) in shaping the large-scale

biospheric drought response, in addition to the drought duration. While droughts in energy-limited regions (aridity index < 1) usually have no or even beneficial impacts, droughts in water-limited



regions (aridity index > 1) have major implications. These contrasting drought impacts imply a critical need for future climate projections to accurately capture regions where the climate can be expected to change from transitional (aridity index ≈ 1) to water-limited (aridity index > 1)

conditions. In such regions, the vegetation drought response will likely become much more pronounced, assuming that the relationships identified here also hold for increased future $CO_2$ levels.



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





**Acknowledgements:**

This study was supported by funding from the German Research Foundation (Emmy Noether grant number 391059971), and from the Swedish Research Council Formas (grant number 2016-02045). We acknowledge the E-OBS dataset from the EU-FP6 project ENSEMBLES (https://www.ecmwf.int/en/research/projects/ensembles, accessed 2 August 2017) and the data

providers in the ECA&D project (http://www.ecad.eu, accessed 2 August 2018), the NASA/GEWEX SRB project (http://gewex-srb.larc.nasa.gov/ accessed on 2 August 2018) and the NASA CERES experiment (http://ceres.larc.nasa.gov/index.php, accessed on 2 August 2018) for sharing radiation data,  and the European Space Agencies' Climate Change Initiative for sharing surface soil moisture data (http://www.esa-soilmoisture-cci.org, accessed on 2 August 2018).

Moreover, we acknowledge all modelling teams contributing simulations to the Earth2Observe and TRENDY projects. Further, we thank Ulrich Weber and Sujan Koirala for help with the model data handling.



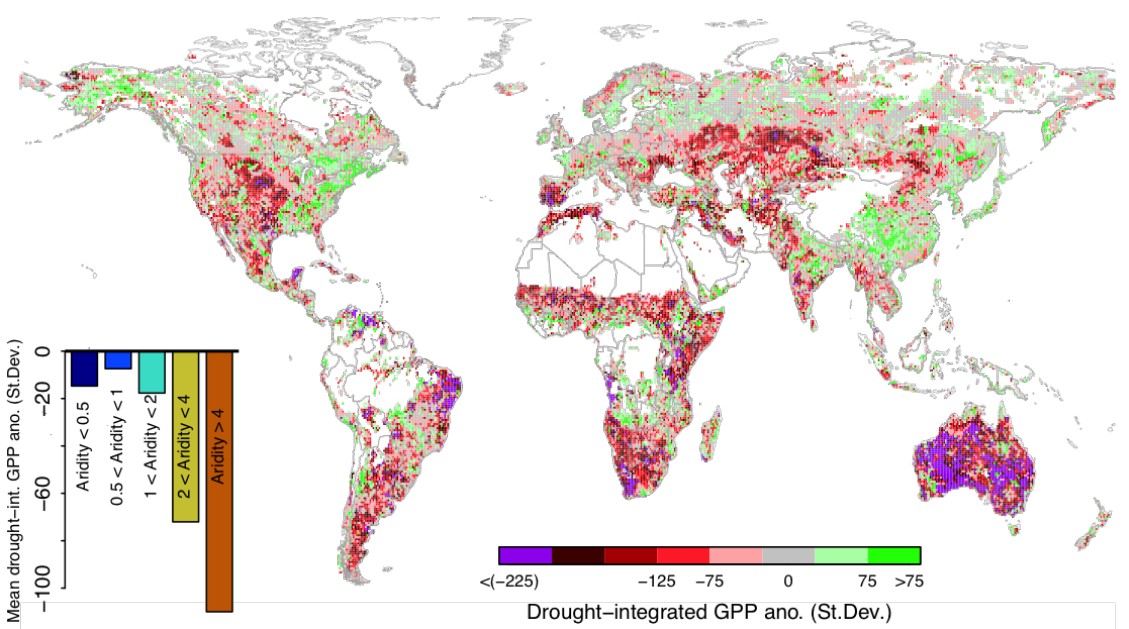

*Figure 1: GPP anomalies integrated over 8 months during the strongest soil moisture drought observed during the study period 2001-2015. Barplot denotes mean anomalies across aridity regions. Regions shown in white have too little soil moisture and/or vegetation data to obtain meaningful results (less than 8 years of at least 50% growing season data availability).*






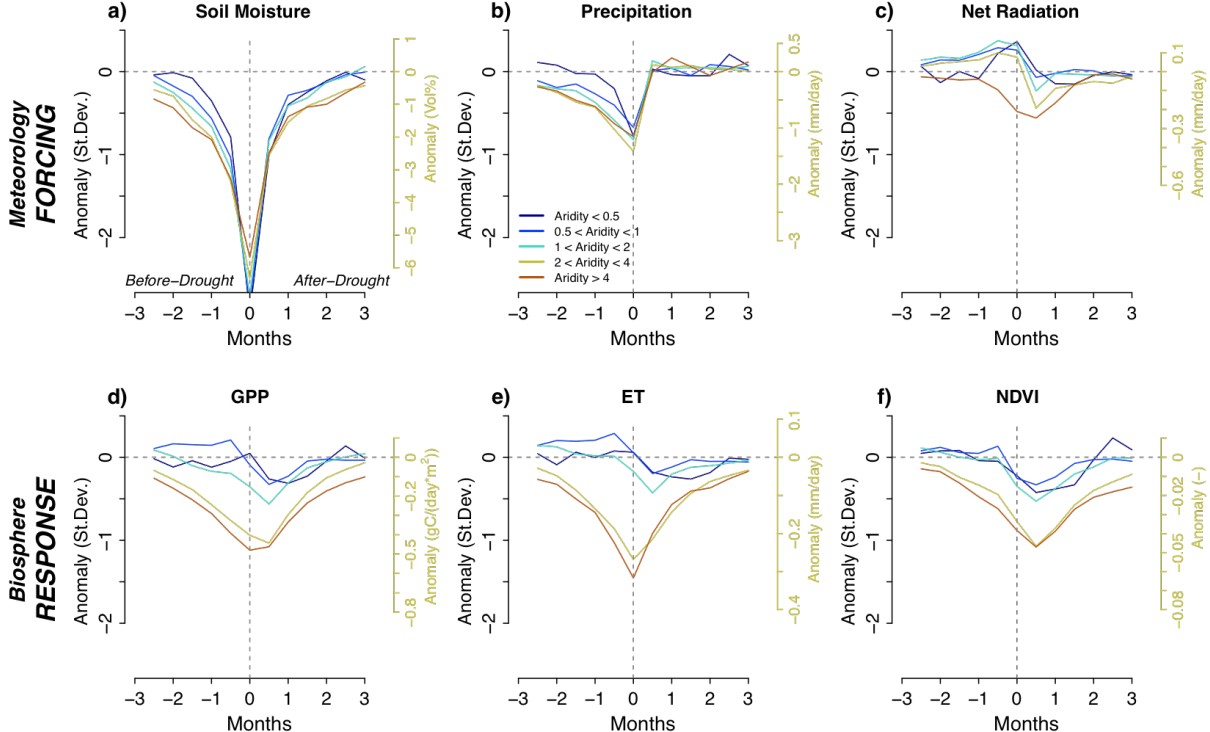

*Figure 2: Aridity-specific time evolution of meteorological forcing (a-c) and biospheric response (d-e) during drought. Evolution for each variable computed as a composite across all grid cells of the respective aridity regions. To ensure comparability of anomalies across variables, values are normalized by and expressed as inter-annual standard deviation of each variable (left axes). Normalization is performed by dividing the actual anomalies (right axes, example for aridity values between 2-4) through the typical aridity-specific variability as expressed by the inter-annual standard deviation across all absolute, half-monthly growing-season anomalies averaged across all grid cells of each aridity region.*



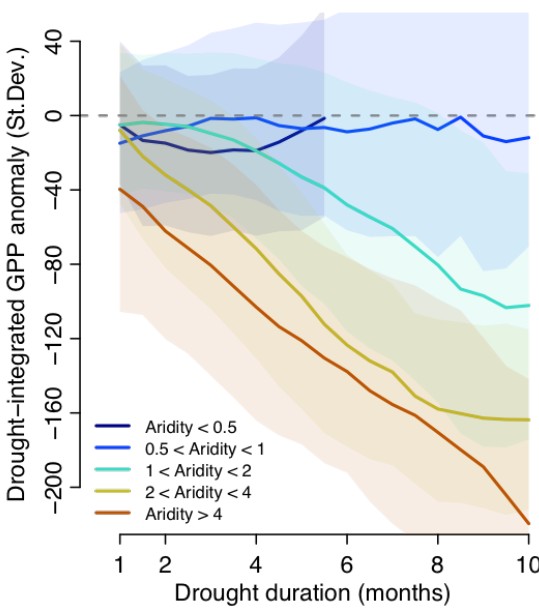

*Figure 3: Drought duration controls integrated biospheric drought response across global aridity regions. Lines obtained through averaging within a 1-month moving window, i.e. GPP anomaly at e.g. 4 months drought duration is inferred with data between 3.5 and 4.5 months drought duration. Lines are computed if more than 50 values are available within moving window. Shadows denote inter-quartile range determined within moving window.*



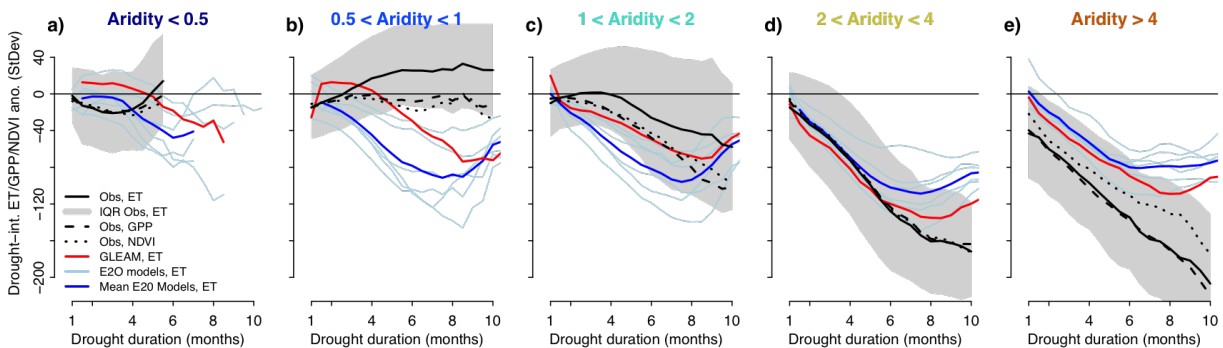

*Figure 4: Drought duration control on biospheric drought response in observations and models.*
*Response of drought-integrated biospheric anomalies across observation-based variables (ET,*
*NDVI, and GPP as displayed in Fig. 3), as well as for modelled ET (GLEAM and Earth2Observe*
*models).*





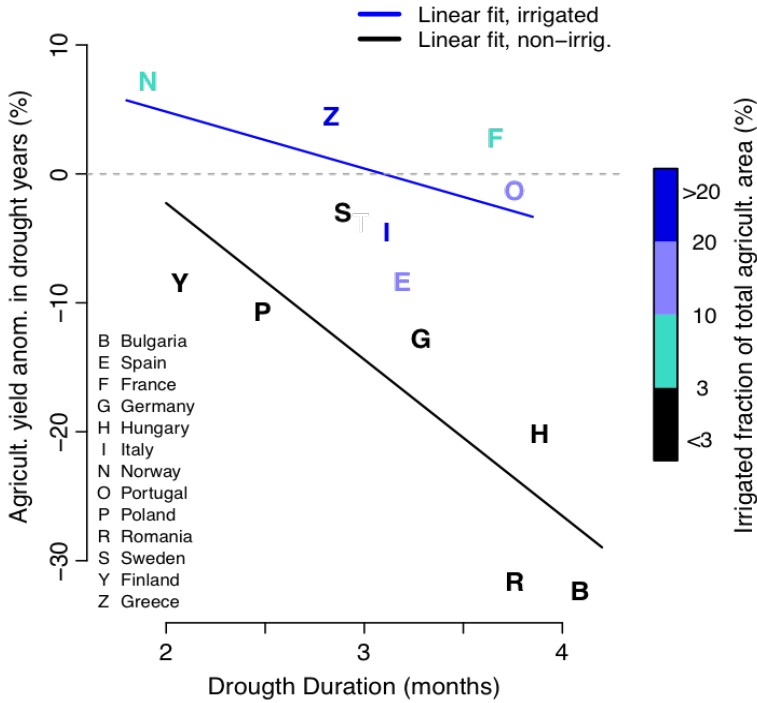

*Figure 5: Drought duration further controls drought-induced yearly agricultural yield anomalies*

*across various European countries. Different line slopes are found for countries where irrigation is*

*applied in agriculture (light blue least-squares fit) than for countries without large-scale*

*agricultural irrigation (gray least-squares fit).*