# Peer review of "Large-scale biospheric drought response intensifies linearly with drought duration in arid regions"

_Biogeosciences, 2019_

## Referee Comment (RC1) · Anonymous Referee #1 · 29 Jan 2020

Review for bg-2019-442 Title: Large-scale biospheric drought response intensifies linearly with drought duration

This study investigated the impacts of soil moisture droughts on several variables (NDVI, GPP, ET and crop yield) to infer the vegetation drought response. And the authors found a linear relationship between drought duration and these variables. The contents are well-organized and the evidences supporting the findings are strong. In general, this paper is already well-written. The reviewer only has a few minor concerns and suggestions for the authors to consider. (1) For GPP and ET, how many machine learning products did you use? It will benefit readers' understanding if you can add a table listing the name, spatial resolution, temporal resolution and temporal coverage of all the datasets used in this study. (2) You categorized machine learning GPP and

ET as observations (Figure 4 and method part), and GLEAM ET as model results. In fact, both machine learning ET and remote sensing based ET are observation-based ET estimates. I suggest you to change the terminology, change "Obs, ET" (Figure. 4a) to "ML, ET". (3) Line 105, what is the unit of the aridity index used in your study? (4) Line 185, how was the power of drought duration calculated? I suggest you to add it in method. (5) Line 187-189, "Other...days." However, according to Figure S6, the explanatory power of "number of dry days" is larger than that of "drought duration". Can you explain it?

---

## Referee Comment (RC2) · Anonymous Referee #2 · 22 Feb 2020

The paper investigates responses of a variety of vegetation characteristics to soil droughts. The authors found the integrated vegetation response may have a linear relationship with drought duration, depending on the background aridity conditions. They further explain such a phenomenon from a water- or energy-limited regime of ecosystems. In general, the paper is well written and organized and I only have a few comments. Therefore, I recommend a minor revision to be warranted. The following lists my concerns or comments: 1) As the authors stated, there is no significant linear correlation between vegetation response and drought duration in wet areas (aridity index < 1), where ecosystems are energy-limited. Meanwhile, across regions of different aridity regimes, the vegetation response to drought duration is not linear as the slope (maybe we can call it intensity of vegetation response) increases towards wet regions.

Thus, I suggest that the title may need revision or more accurate delimitation because its current form is somewhat misleading. 2) Line 95: since all anomalies are scaled by standard deviation, how could they be still expressed as inter-annual standard deviations? It seems the authors are actually using z-score. Moreover, could the authors add some formulas for their computation? It will be helpful for readers to understand the data processing. 3) Lines 116-118: references are required.

---

## Author Comment (AC1) · 6 Mar 2020

Review for bg-2019-442 Title: Large-scale biospheric drought response intensifies linearly with drought duration This study investigated the impacts of soil moisture droughts on several variables (NDVI, GPP, ET and crop yield) to infer the vegetation drought response. And the authors found a linear relationship between drought duration and these variables. The contents are well-organized and the evidences supporting the findings are strong. In general, this paper is already well-written.

A1: We thank the reviewer for this encouraging evaluation.

The reviewer only has a few minor concerns and suggestions for the authors to consider. (1) For GPP and ET, how many machine learning products did you use? It will

benefit readers' understanding if you can add a table listing the name, spatial resolution, temporal resolution and temporal coverage of all the datasets used in this study.

A2: We thank the reviewer for this suggestion, and have implemented it by introducing a respective table into the manuscript in line 495.

(2) You categorized machine learning GPP and ET as observations (Figure 4 and method part), and GLEAM ET as model results. In fact, both machine learning ET and remote sensing based ET are observation-based ET estimates. I suggest you to change the terminology, change "Obs, ET" (Figure. 4a) to "ML, ET".

A3: We agree with the reviewer and have replaced 'Obs' with 'Reference' within Figure 4. Further, we have updated the caption of Figure 4 in lines 515-519:

"Figure 4: Drought duration control on biospheric drought response in observations and models. Response of drought-integrated biospheric anomalies across observation-based reference data (ET, NDVI, and GPP as displayed in Fig. 3), as well as for modelled ET (GLEAM and Earth2Observe models)."

(3) Line 105, what is the unit of the aridity index used in your study?

A4: It is unitless, and we clarify this point in lines 105-107:

"This unitless index is derived by dividing the mean net radiation over the entire study period by the corresponding unit-adjusted precipitation mean."

(4) Line 185, how was the power of drought duration calculated? I suggest you to add it in method.

A5: We have added a sentence on this in lines 187-189:

"It is computed as the correlation between the drought-integrated GPP anomalies and the respective drought metric values from the droughts across all grid cells of a particular aridity class."

(5) Line 187-189, "Other. . .days." However, according to Figure S6, the explanatory power of "number of dry days" is larger than that of "drought duration". Can you explain it?

A6: We added a sentence to clarify this point in lines 195-197:

"Only the number of dry days (within the soil moisture-diagnosed drought duration) yields slightly higher correlations as in the case of drought duration, which results from the additional, precipitation-based information contained in the number of dry days."

---

## Author Comment (AC2) · 6 Mar 2020

The paper investigates responses of a variety of vegetation characteristics to soil droughts. The authors found the integrated vegetation response may have a linear relationship with drought duration, depending on the background aridity conditions. They further explain such a phenomenon from a water- or energy-limited regime of ecosystems. In general, the paper is well written and organized and I only have a few comments. Therefore, I recommend a minor revision to be warranted.

B1: We thank the reviewer for these positive comments.

The following lists my concerns or comments: 1) As the authors stated, there is no significant linear correlation between vegetation response and drought duration in wet

areas (aridity index < 1), where ecosystems are energy-limited. Meanwhile, across regions of different aridity regimes, the vegetation response to drought duration is not linear as the slope (maybe we can call it intensity of vegetation response) increases towards wet regions. Thus, I suggest that the title may need revision or more accurate delimitation because its current form is somewhat misleading.

B2: We agree with the reviewer, and have added 'in arid regions' to the title.

2) Line 95: since all anomalies are scaled by standard deviation, how could they be still expressed as inter-annual standard deviations? It seems the authors are actually using z-score. Moreover, could the authors add some formulas for their computation? It will be helpful for readers to understand the data processing.

B3: The reviewer is correct, and we have clarified this point in lines 92-97:

"To enable direct comparison of anomalies across variables, and across observations and models, we compute z-scores. This is done by standardizing all anomaly values by dividing them with a characteristic variability value. This value is computed for each variable and each grid cell as the standard deviation across all half-monthly growing-season values. This way, all anomalies discussed and illustrated in this study are scaled by inter-annual standard deviations to be expressed as z-scores."

3) Lines 116-118: references are required.

B4: We have added a reference to this statement in line 119.

---

## Author Response (AR1)

**Editor comments**

A1: We thank the editor for the quick handling of our manuscript in these difficult times.

Comments to the Author:
Based on the reviewers comments two clarification are asked:
1. As a common aridity index is based on the ratio between the annual precipitation to potential evapotranspiration, the authors are asked to refer here to Budyko (or a newer reference) and state that L is the latent heat of water vaporization.

A2: We have added the reference to the Budyko paper in lines 105-109:

"This index was originally introduced as the ratio between mean potential evapotranspiration over the study period and the respective mean precipitation, with the latter scaled by the latent heat of vaporization to yield a unitless index value *(25)*. We use an adapted form where we replace the potential evapotranspiration with satellite-derived net radiation."

2. Please clarify the answer for the comment on line 185 as in the Authors answer at lines 187-9.

A3: We have adapted the respective paragraph in lines 190-194:

"It is computed by (i) obtaining the drought-related GPP anomaly accumulated over an 8-month time window containing the drought period (see methods), and the respective drought metric values for each grid cell and its respective strongest drought, and (ii) calculating the correlation between the drought-related GPP anomalies and the respective drought metric values across all grid cells of each aridity class."